

# A closer look at the taxonomic and genetic diversity of endemic South African *Marphysa* Quatrefages, 1865

Jyothi Kara[1,2,*], Isabel C. Molina-Acevedo[3,*], Angus Macdonald[4], Joana Zanol[5] and Carol Simon[6]

[1] Research and Exihibitions, Iziko South African Museums, Cape Town, Western Cape, South Africa
[2] Conservation and Marine Science, Cape Peninsula University of Technology, Cape Town, South Africa
[3] Centro Regional de Investigación Acuícola y Pesquera, Manzanillo, Colima, Mexico
[4] Biological Sciences, University of KwaZulu-Natal, Durban, KwaZulu-Natal, South Africa
[5] Department of Invertebrates, Universidade Federal do Rio de Janeiro, São Cristovão, Brazil
[6] Botany and Zoology, University of Stellenbosch, Stellenbosch, Western Cape, South Africa
* These authors contributed equally to this work.

Corresponding authors
Jyothi Kara, jyothi.kara@gmail.com
Isabel C. Molina-Acevedo, isacrismoliace@gmail.com

## ABSTRACT

The current study investigates the final unresolved cosmopolitan species of *Marphysa* in South Africa, *Marphysa corallina*, collected from KwaZulu Natal, Eastern and Western Cape provinces, together with another species collected from northern KwaZulu Natal. Morphological and genetic data prove that *M. corallina*, originally described from Hawaii, does not occur in South Africa. The curvature of the inner base on maxilla I, the elevated inner base of maxilla II, and the ventral cirrus as a transverse welt with a rounded tip allow us to identify it as a new species of *Treadwellphysa, T. izinqa* sp. nov. (common name: brown wonderworm). Characteristic traits include the basal reddish and distal golden colour of the subacicular hook, the ear-shaped postchaetal lobe, and tridentate falcigers which is reported for the first time for the genus. This species is harvested as bait on the south coast of SA, although less frequently than the more common blood wonderworm, *Marphysa haemasona* Quatrefages, 1866, and can be distinguished by its more uniform brown colouration and white-tipped antennae. A second species, *Marphysa mzingazia* sp. nov., is characterized by red eyes, six branchial filaments extending to the posterior end, the golden aciculae in posterior chaetigers, weakly bidentate yellow/brown subacicular hooks, and the presence of similar sized spinigers along the body. A molecular analysis based on cytochrome oxidase I fragments confirm both taxa as different species. A key for all South African species of *Marphysa* is included.

# INTRODUCTION

Until the 1900s polychaete descriptions were often vague, included poor illustrations and lacked diagnostic characters that are currently regarded as important for generic and species level designations (*Hutchings & Kupriyanova, 2018*; *Simon et al., 2022*). Consequently, there was a bias toward using easy-to-identify, obvious characters.
For example, the presence, shape, and distribution of branchiae were considered diagnostic for generic delimitation in Eunicidae Berthold, 1827, while today, these characters are found to vary within species and are deemed uninformative at the genus level (*Carrera-Parra & Salazar-Vallejo, 1998*; *Zanol, Halanych & Fauchald, 2014*). Revisions of species based on type and non-type material and the use of molecular data are increasingly important in standardizing key characters for species delimitation (*Molina-Acevedo & Idris, 2020*, *2021*; *Zanol et al., 2021*). This was evident for the genus *Marphysa* Quatrefages, 1865, which since the designation of neotype material for the type species of the genus in 2003, has seen an explosion in the number of incorrect synonymizations reversed and new species described (*Fauchald, 1970*; *Hutchings & Karageorgopolous, 2003*; *Glasby & Hutchings, 2010*; *Idris, Hutchings & Arshad, 2014*; *Zanol, Halanych & Fauchald, 2014*; *Molina-Acevedo & Carrera-Parra, 2015*; *Zanol, da Silva & Hutchings, 2017*; *Read & Fauchald, 2023*).

In southern Africa, *Marphysa* was ranked 7[th] on the list of prioritised genera for taxonomic revision and this probably applies to records from the east and west African regions (*Simon et al., 2021a*, *2022*; *Read & Fauchald, 2023*). It includes several species from this genus that are used as bait for recreational and subsistence fishing in the region and although different taxa are used, the inconsistent use of common names complicates the identification and hinders appropriate management strategies. For example, fisherman use six common names for two *Marphysa* species, while the most widely used name, *i.e.*, wonder worm also refers to species of *Eunice* and *Lysidice* (*Simon et al., 2021a*, *2021b*, *2022*). Correctly identifying species and clarifying indigenous diversity is therefore imperative to facilitate their management and to avoid over-exploitation of resources (*Simon et al., 2019*, *2021a*, *2021b*).

Of the seven *Marphysa* species recorded for southern Africa in *Day (1967)*, only three were considered endemic. The remaining four had type localities outside the region and were considered unresolved cosmopolitan species (*sensu Darling & Carlton, 2018*). Nonetheless, it was found that *Marphysa haemasona* Quatrefages, 1866 (previously *Marphysa sanguinea*, *M. elityeni Lewis & Karageorgopoulos, 2008*) and *Marphysa durbanensis* Day, 1934 (previously *Marphysa macintoshi* Crossland, 1903) were incorrectly synonymised endemic species and thus were reinstated (*Lewis & Karageorgopoulos, 2008*; *Kara et al., 2020b*). In the last few years, two of the three indigenous species recorded were moved to two newly erected genera, *Nicidion posterobranchia* (Day, 1962) (previously *Marphysa posterobranchia*) and *Paucibranchia purcellana* (Willey, 1904) (previously *Marphysa purcellana*) (*Molina-Acevedo, 2018*; *Molina-Acevedo & Idris, 2021*). Lastly, one species, new to science, was described for the region, *Marphysa sherlockae* Kara, Molina-Acevedo, Zanol, Simon & Idris, 2020, as it was incorrectly identified as *Marphysa depressa* (Schmarda, 1861) (*Kara et al., 2020b*). Accordingly, five (instead of seven) species of *Marphysa* are currently known to occur in the region, four of which are valid endemic species, and this study explores the validity of the last unresolved cosmopolitan species recorded for South African waters.

*Marphysa corallina* (Kinberg, 1865) was described from Honolulu, Hawaii and has since been recorded from several disjunct localities including Mozambique, Madagascar,

Senegal, New Zealand, the Red Sea, Australia, Marshall Islands, Lakshadweep Island and the Jaluit Atoll (*Day, 1967*; *Veeramuthu et al., 2012*; *Read & Fauchald, 2023*). In South Africa, this species was recorded from Sodwana Bay (KwaZulu-Natal) to East London (Eastern Cape) on the east coast and an isolated population in Witsand (Western Cape) on the south coast, where it is used as bait (*Kara, 2015*; *Simon et al., 2019*, *2021a*, *2021b*). Specimens in South Africa were identified as *M. corallina* due to the distribution of branchiae, and the presence of compound falcigers and bidentate subacicular hooks (*Day, 1953*, *1967*). However, its presence has not been confirmed using thorough morphological and genetic analyses, and most likely represents a misidentification as demonstrated for the other *Marphysa* species mentioned above. South African *M. corallina* are superficially very similar to *M. haemasona*, a species commonly used as bait. Therefore, identifying characters to distinguish them in the field is important for management. It is imperative that guidelines are provided (*i.e.*, designate a common name and identify features that separate similar species in the field) so that their usage as a bait resource can be documented and their over-exploitation prevented as either species can be mistaken for each other.

In this study we investigated whether *M. corallina* occurs in South Africa by examining the type material from Hawaii and specimens from throughout its known distribution in South Africa, including molecular comparisons where possible. Our results indicate that *M. corallina* is not present in South Africa and instead represents a new species of *Treadwellphysa Molina-Acevedo & Carrera-Parra, 2017*. Finally, we provide an updated description of *M. corallina* using the holotype, a description of a *Marphysa* species new to science and a taxonomic key to all *Marphysa* species from South Africa.

## MATERIALS AND METHODS

### Sample collection

Specimens were collected at the fringing intertidal zones from eight open-coast sites along the KwaZulu-Natal and Eastern Cape coasts from 2013 to 2014 ($n = 32$) and in 2019 ($n = 29$), including an isolated population from rock crevices on the muddy banks of the Mzingazi canal in Richards Bay (Fig. 1). Whole specimens were preserved in 96% ethanol. Collection was approved by the Department of Forestry, Fisheries and the Environment in South Africa under permit numbers RES2013/13, RES2014/06 issued to Angus Macdonald and RES2019/49 issued to Carol Simon.

### Morphological examination

Type material of *Marphysa corallina*, specimens identified as *M. corallina* deposited at the Iziko South African Museum and newly collected specimens were examined and the following characters recorded. Shape of the prostomium, peristomium, anterior body region, maxillary apparatus, parapodia and pygidium, as well as the frequency of branchiae with total number of filaments and the maxillary formula (MF) were recorded. Maxillae I to V (MI, MII, MIII, MIV, MV) were measured according to *Molina-Acevedo & Carrera-Parra (2015*, *2017)*. A Leica S9i dissecting and light ICC50W microscopes equipped with built-in cameras were used to photograph characters and plates were edited

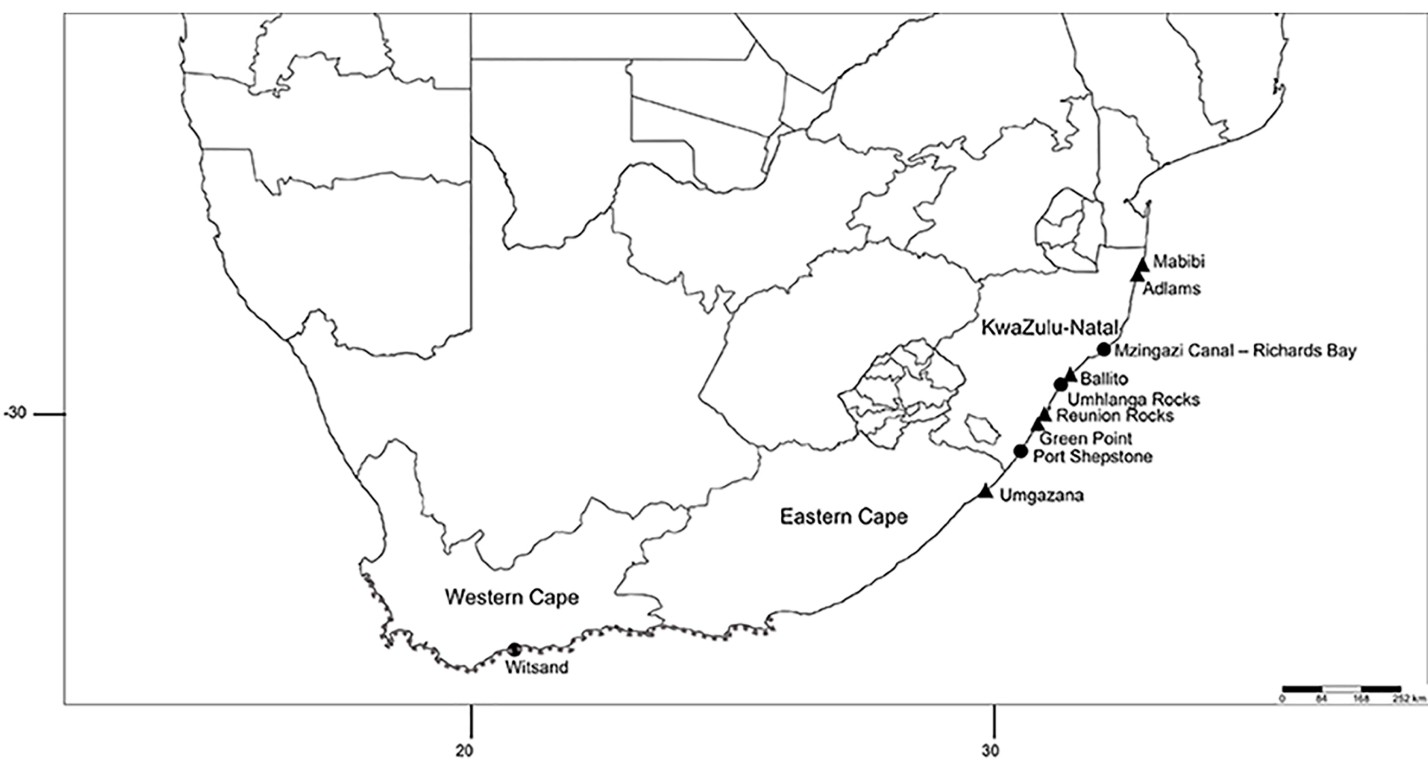

**Figure 1** **Sampling localities of *Treadwellphysa izinqa* sp. nov., *Marphysa mzingazia* sp. nov. and *Marphysa haemasona* in South Africa.** *Treadwellphysa izinqa* sp. nov.: Mabibi, Adlams, Ballito, Umhlanga Rocks, Reunion Rocks, Green Point, Port Shepstone, Umgazana and Witsand and *Marphysa mzingazia* sp. nov.: Mzingazi Canal, Richards Bay. Material collected in 2013 and 2014 are denoted by triangles and those collected in 2019 by circles. Discontinuous dark grey dots: known distribution of *Marphysa haemasona* (*Kara et al., 2020b*; *Simon et al., 2021b*).

in Adobe Photoshop 2022 and included in descriptions. Specimens collected in 2013, 2014 and 2019 were deposited at the Iziko South African Museum (MB-A095266–MB-A095297). Comparative material was loaned from the Swedish Museum of Natural History, Stockholm, Sweden (SMNH) and Iziko South African Museum (MB).

## Molecular analyses

DNA was extracted from tissue samples of fresh specimens using the ZR Genomic DNA Tissue MiniPrep Plus kit (Zymogen) following the manufacturer's protocol. DNA was amplified using universal COI primers LCO1490 and HCO2198 (*Folmer et al., 1994*). Polymerase Chain Reaction (PCR) amplifications followed *Kara et al. (2020b)*. Amplicons were sequenced at the Central Analytical Facility at Stellenbosch University using only the forward primer (LCO1490). Raw sequences were quality controlled to check for errors using BioEdit (v7.2.6) (*Hall, 1999*).

GenBank sequences of species belonging to *Marphysa, Paucibranchia Molina-Acevedo, 2018*, *Eunice* Cuvier, 1817, *Palola* Gray in Stair, 1847 and *Leodice* Lamarck, 1818 were included as ingroup taxa together with 32 newly generated sequences in this study (GenBank accession numbers: OQ836443–OQ836473, whilst *Hyalinoecia* Malmgren,

**Table 1** Sequence data used in the phylogenetic analysis.

| Species | Genbank accession number | Type locality | Collection locality | Reference |
|---|---|---|---|---|
| *Marphysa mzingazia* sp. nov. | OQ836468–OQ836473 | KwaZulu-Natal, South Africa | Mzingazi Canal, Richards Bay, South Africa | This study |
| *Treadwellphysa izinqa* sp. nov. | OQ836443–OQ836467 | KwaZulu-Natal, South Africa | Umhlanga Rocks, Port Shepstone, South Africa | This study |
| *Treadwellphysa izinqa* sp. nov. (=*Marphysa corallina*) | KT823371, KT823360, KT823349, KT823372, KT823356, KT823389, KT823379, KT823380, KT823388, KT823386, KT823387, KT823343, KT823330, KT823328, KT823329, KT823342, KT823306, KT823309, KT823311, KT823312, KT823314, KT823300, KT823301, KT823288, KT823294, KT823300, KT823397, KT823398, KT823402, KT823407, KT823410, MN067881, MN067882 | KwaZulu-Natal and Western Cape, South Africa | Mabibi, Adlams, Ballito, Reunion Rocks, Green Point, Umgazana, Witsand | *Kara (2015)*, *Simon et al. (2021b)* |
| *Marphysa haemasona* | MN067877 | Cape of Good Hope, South Africa | Kommetjie, South Africa | *Simon et al. (2021b)* |
| *Marphysa sherlockae* | MT840349–MT840351 | Durban, South Africa | Strand, South Africa | *Kara et al. (2020b)* |
| *Marphysa aegypti* | MF196971, MF196969, MF196970, MF196968 | Suez Canal, Egypt | Suez Canal, Egypt | *Elgetany et al. (2018)* |
| *Marphysa chirigota* | MN816441, MN816442, MN816443 | Cádiz Bay, SW Iberian Península | Cádiz Bay, SW Iberian Península | *Martin et al. (2020)* |
| *Marphysa bifurcata* | KX172177, KX172178 | Sheltered North Reef at Poin Peron, Western Australia | Australia | *Zanol, da Silva & Hutchings (2016)* |
| *Marphysa brevitentaculata* | GQ497548 | Scarborough, Tobago, Trinidad and Tobago | Mexico | *Zanol et al. (2010)* |
| *Marphysa californica* | GQ497552 | San Diego County, California | California | *Zanol et al. (2010)* |
| *Marphysa fauchaldi* | KX172165 | off Elizabeth River, Darwin region, Australia | Australia | *Zanol, da Silva & Hutchings (2016)* |
| *Marphysa gaditana* | MN816444, KR916870, AY040708, KR916871, KR916872, KR91687, KP254503, KP254537, KP254643, KP254743, KP254802 | Cádiz Bay, SW Iberian Península | Cádiz Bay, SW Iberian Península Portugal, France, Virginia (USA) | *Martin et al. (2020)*, *Lobo et al. (2016)*, *Siddall et al. (2014)*, *Leray & Knowlton (2015)* |
| *Marphysa honkongensa* | MH598526 | Tolo Harbour, Hong Kong | China | *Wang, Zhang & Qiu (2018)* |
| *Marphysa iloiloensis* | MN133418, MN106279, MN106280, MN106281 | Tigbauan, Iloilo Province | Philippines | *Glasby et al. (2019)* |
| *Marphysa kristiani* | KX172159, KX172160, KX172161, KX172162, KX172163 | Stingray Bay, New South Wales | Australia | *Zanol, da Silva & Hutchings (2016)* |
| *Marphysa mossambica* | JX559751, KX172164 | Mozambique | Philippines, Australia | *Zanol et al. (2010)*, *Zanol, da Silva & Hutchings (2016)* |

(Continued)

| | | | | |
|---|---|---|---|---|
| **Table 1 (continued)** | | | | |
| **Species** | **Genbank accession number** | **Type locality** | **Collection locality** | **Reference** |
| *Marphysa mullawa* | KX172166, KX172168, KX172174, KX172175, KX172176 | Moreton Bay, Fisherman's, Queensland | Australia | *Zanol, da Silva & Hutchings (2016)* |
| *Marphysa pseudosessiloa* | KY605405, KY605406 | Careel Bay, New South Wales | Australia | *Zanol, da Silva & Hutchings (2017)* |
| *Marphysa victori* | MG384996, MG384999, MG384997, MG384998 | Arcachon Bay | France | *Lavesque et al. (2017)* |
| *Marphysa viridis* | GQ497553 | Boca Grande Key, Florida | Brazil | *Zanol et al. (2010)* |
| *Marphysa sanguinea* | GQ497547, MK541904, MK950851, MK950852, MK950853, MK967470, MN106282, MN106283, MN106284 | Polperro, Cornwall | Cornwall (UK), France | *Zanol et al. (2010), Lavesque et al. (2019), Glasby et al. (2019)* |
| *Marphysa tripectinata* | MN106271, MN106272, MN1062723, MN106274, MN106275, MN106276, MN106277, MN106278 | Beihai, China | China | *Liu, Hutchings & Kupriyanova (2018)* |
| *Paucibranchia bellii* | KT307661 | Chausey Island, France | Spain | *Aylagas et al. (2016)* |
| *Paucibranchia disjuncta* | GQ497549 | Los Angeles County, California | California, USA | *Zanol et al. (2010)* |
| *Palola viridis* | GQ497556 | Samoa | Micronesia | *Zanol et al. (2010)* |
| *Eunice cf. violaceomaculata* | GQ497542 | – | Belize | *Zanol et al. (2010)* |
| *Leodice rubra* | GQ497528 | – | Brazil | *Zanol et al. (2010)* |
| *Hyalinoecia* sp. | GQ497524 | – | Massachusetts, USA | *Zanol et al. (2010)* |

**Note:**
GenBank accession numbers, type locality, collection locality and references of mitochondrial cytochrome c oxidase subunit 1 (COI) sequences of taxa used in the molecular analyses.

1867 was used as the outgroup to root the tree (Table 1). The COI dataset was trimmed and aligned using the ClustalW multiple alignment method (*Thompson, Higgins & Gibson, 1994*) in BioEdit. A nexus file was compiled using Dna-SP v5 (*Librado & Rozas, 2009*). A best-fit evolution model was calculated using PAUP (*Swofford, 2003*) and MrModelTest v2.3 (*Nylander, 2004*). Using the Akaike Information Criterion (AIC), the SYM+G model was used to reconstruct phylogenetic relationships using Bayesian Inference (BI) implemented in MrBayes 3.1.2 (*Ronquist et al., 2012*). Trees were calculated using four Markov Chains of 5 million generations with every 1,000[th] tree sampled. The first 25% were discarded as burn-in and the resulting trees used to build a 50% majority-rule consensus tree. Convergence of runs was assessed through examining the average standard deviation of split frequencies (≤0.01). The plot of likelihood *vs* the sampled trees and the effective sample sizes (ESS > 200) were analysed using Tracer v1.5 (*Rambaut, 2012*) to verify the mixing quality of all parameters, both of which were satisfied. Trees were visualised using FigTree v1.4.4 (*Rambaut, 2012*) and edited in Photoshop. Our analysis was designed to validate species identities and not to assess their phylogenetic relationships.

Inter- and intraspecific genetic distances were computed using the Kimura 2-parameter model (K2P) in MEGA-X (*Kumar et al., 2018*) and run for 100,000 bootstrap replicates with complete deletion of gaps.

# RESULTS

## Morphological and molecular comparisons

Detailed morphological comparisons indicate that *M. corallina* was misidentified in the region, and instead, the specimens correspond to a new species belonging to *Treadwellphysa Molina-Acevedo & Carrera-Parra, 2017*, and represents the first record for South Africa. Diagnostic characters for this genus are the curvature of the inner base on MI and the elevated inner base of MII together with the ventral cirrus as a transverse welt with a rounded tip (*Molina-Acevedo & Carrera-Parra, 2017*). These specimens belong to a new species, herein named *T. izinqa* sp. nov. characterised by having reddish/golden subacicular hooks, a postchaetal lobe varying from digitiform to ear-shaped to inconspicuous, and tridentate falcigers (unique for the genus) of consistent length throughout the body.

COI sequences from Hawaiian *M. corallina* were not available. Nonetheless, South African sequences formed a distinct clade (intraspecific variation: <1%, BS > 0.95), differing genetically from other sequenced species of *Marphysa* by 21–25% (Fig. 2).

Our samples revealed another distinct morpho-species, characterised by having red eyes, branchiae with six filaments extending to the posterior end, aciculae varying from black/amber to golden, yellow or brown/golden subacicular hooks blunt or weakly bidentate, and spinigers of similar size all along the body. Accordingly, their sequences correspond to an independent clade (intraspecific variation: <1%, BS > 0.95) (Fig. 2) differing genetically from other sequenced *Marphysa* by 9–21%. Thus, the species was herein named *Marphysa mzingazia* sp. nov.

## Systematics

**Eunicida Dales, 1962**

**Eunicidae Berthold, 1827**

**Genus *Marphysa* de Quatrefages, 1865**

***Marphysa corallina* (Kinberg, 1865)**

(Fig. 3)

*Material examined.* Holotype: one specimen in poor condition, SMNH-Type-429, Hawaiian Islands, Oahu, Honolulu, 21°19′N, 157°52′W, Eugenie Epx. 1851–53 (three vials each with parapodia, and one vial with the maxillary apparatus).

*Redescription.* Holotype incomplete, with 104 chaetigers, (fragments: anterior = 9 chaetigers, second and third = 19 each, fourth = 2, fifth = 70, sixth = 3), L10 = 6.1 mm, W10 = 3.4 mm. Anterior region with dorsum convex, flat ventrally widest at chaetiger 13, tapering after chaetiger 48. Prostomium bilobed (0.9 mm long, 2.4 mm wide), sulcus anteriorly shallow, dorsally inconspicuous and ventrally deep (Figs. 3A and 3B). Prostomial appendages in a semicircle, median antenna slightly isolated by a gap. Palps reaching second peristomial ring; lateral antennae reaching first chaetiger; median antenna
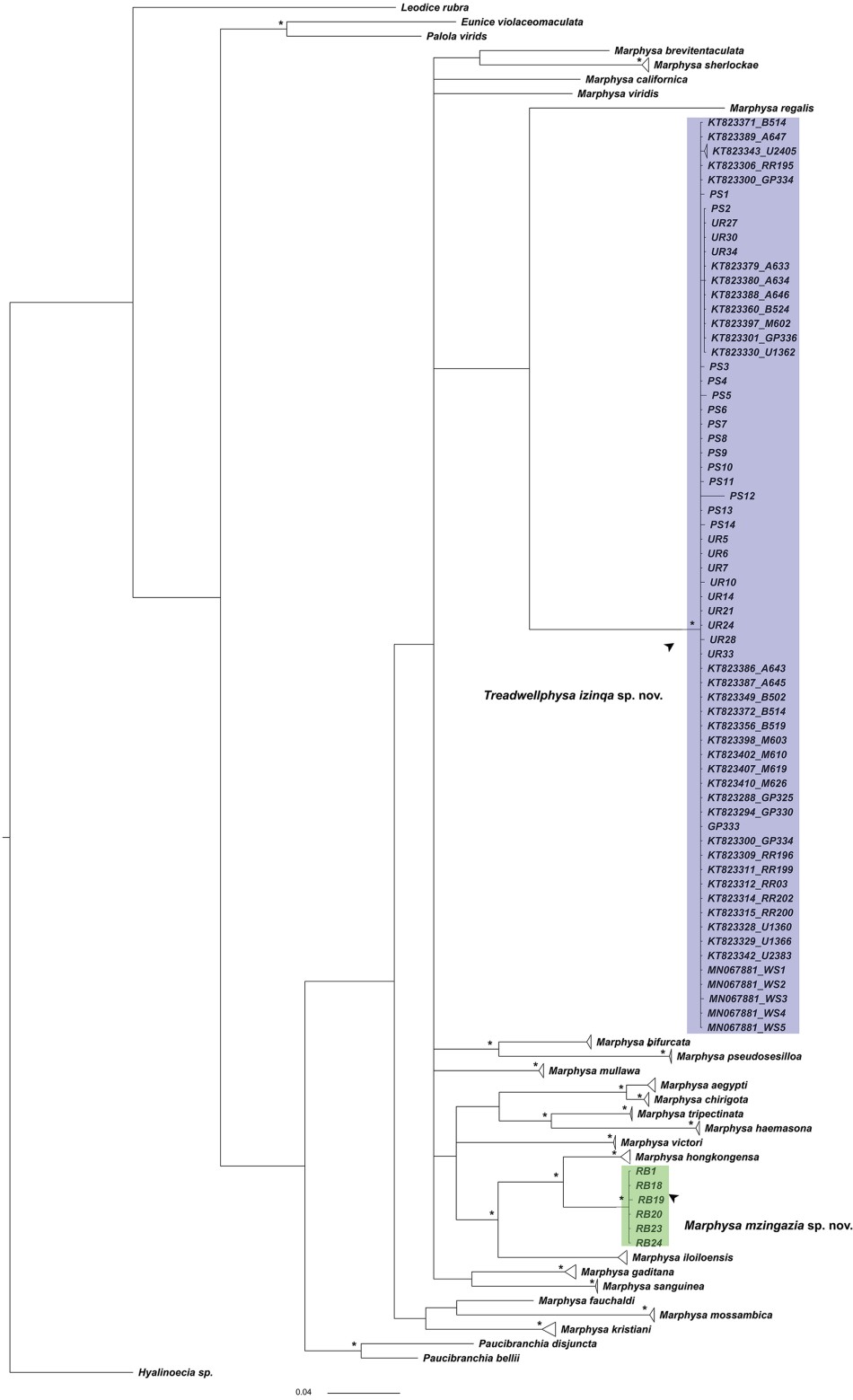

**Figure 2** **Bayesian tree based on mitochondrial cytochrome c oxidase subunit 1 sequences of *Marphysa* spp. and *Treadwellphysa*.** *Bayesian posterior probabilities >95%, purple clade: *Treadwellphysa izinqa* sp. nov. and green clade: *Marphysa mzingazia* sp. nov.; scalebar: the number of substitutions per site.

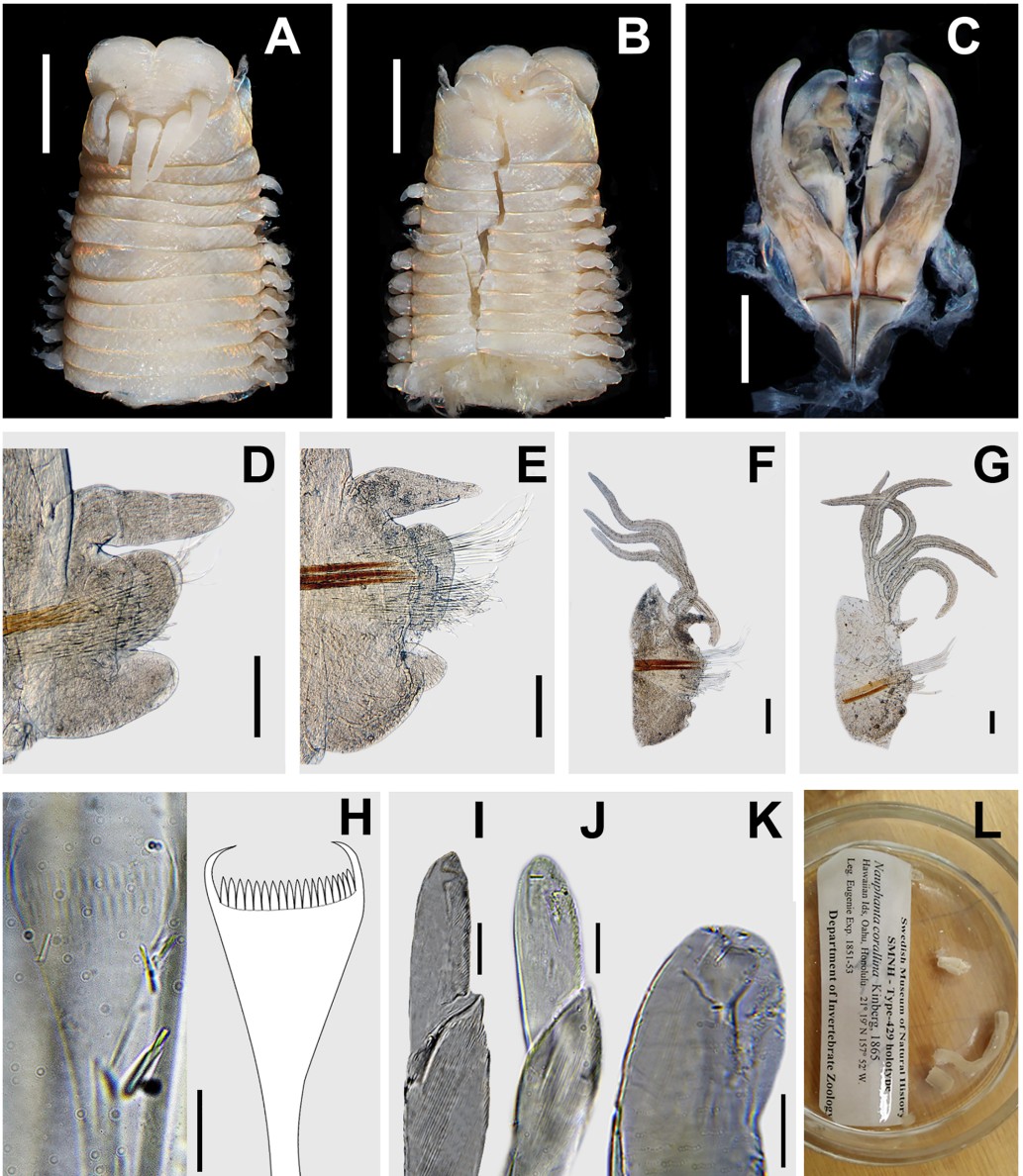

**Figure 3 *Marphysa corallina* (Kinberg, 1865) holotype SMNH-type-429.** (A) Anterior end, dorsal view; (B) anterior end, ventral view; (C) maxillary apparatus, dorsal view; (D) parapodia from chaetiger 3; (E) parapodia from chaetiger 14; (F) parapodia from chaetiger 54; (G) parapodia from chaetiger 98; (H) isodont pectinate chaeta with wide blade and short, slender teeth, chaetiger 54; (I) isodont pectinate chaeta with narrow blade and short, slender teeth, chaetiger 98; (J) compound falciger, chaetiger 14; (K) compound falciger chaetae, chaetiger 98; (L) subacicular hook, chaetiger 54. All chaetiger in anterior view. Scale bars: A, B, = 1.5 mm; C = 0.8 mm; D–F = 0.3 mm; G = 0.25 mm; H = 10 μm; I = 8.5 μm; J, L = 12.3 μm.

reaching second chaetiger (Fig. 3A). Palpophores and ceratophores short, thick; palpostyles and ceratostyles digitiform, slender, without articulations (Figs. 3A and 3B). Eyes brown, between palps and lateral antennae.

Peristomium (1.5 mm long, 3.6 mm wide) with first ring two times longer than second ring. Separation between rings distinct on all sides. Ventral lip dissected (Figs. 3A and 3B).

Maxillary apparatus in poor condition (Fig. 3C), with MIII, MIV and MV lost; MF = 1+1,4+4, ?+? ?+?, ?+? MI with falcal arch angular shaped and with straight outer edge (Fig. 3C). MI forceps-like, 2.5 times longer than length of maxillary carriers. MII wide, left distal teeth shorter, directed laterally, other teeth recurved; cavity opening oval, MII 3.8 times longer than length of cavity opening (Fig. 3C).

Pectinate branchiae from chaetigers 17L–18R to last chaetiger (Figs. 3F and 3G), with up to five long branchial filaments 5.3 times longer than dorsal cirri.

Dorsal cirri digitiform in anterior chaetigers, conical in following ones; 1.1 times longer than ventral cirri in first chaetigers, similar in length to ventral cirri in median-posterior ones (Figs. 3D–3G). Ventral cirri tongue-shaped in first four chaetigers; with an elongated oval swollen base from chaetiger 5. Prechaetal lobes in first chaetigers with dorsal edge 1.3 times longer than ventral, as transverse fold in following chaetigers (Figs. 3D–3G); chaetal lobes rounded in anterior chaetigers, triangular in following ones (Figs. 3D–3G); postchaetal lobes well developed in first 58 chaetigers; tongue-shaped in first four chaetigers, ear-shaped from chaetiger 5, progressively smaller in following ones (Figs. 3D–3G).

Aciculae blunt, reddish basally, amber distally, up to three aciculae per parapodia (Figs. 3D–3G). Limbate chaetae of two lengths in same chaetiger. Two types of pectinate chaetae: 2–3 isodont pectinate chaetae with wide blade, thin shaft, and up to 11 long, slender teeth (Fig. 3H) in anterior chaetigers; 2–3 pectinate chaetae isodont with narrow blade, thin shaft, and with up to 20–21 short, slender teeth (Fig. 3I) in median chaetigers. Pectinate chaetae from posterior region not observed. Compound falcigers bidentate, with blade of similar lengths in all parapodia, with triangular teeth in anterior region (Fig. 3J) and blunt teeth in posterior parapodia (Fig. 3K). Subacicular hooks starting in chaetigers 30R–32L, present in all parapodia throughout the body, always 2–3 per parapodium; bidentate and translucent with blunt teeth, similar in size (Fig. 3L).

**Distribution.** Oahu, Hawaii. Specimen records from other regions outside the type locality such as Madagascar, Mozambique, Senegal, the Red Sea, the Mediterranean, Jaluit Athol, Lakshadweep Islands, Australia and New Zealand (*Day, 1967*; *Branch et al., 2022*; *Veeramuthu et al., 2012*; *Read & Fauchald, 2023*) should be re-examined based on the updated description as it is suspected that these records may represent misidentified species.

***Marphysa mzingazia* sp. nov.**
lsid:zoobank.org:act:6A50F2CF-2DE3-42AC-8B4D-FECC6C04D4BF
(Fig. 4)

**Material examined**. *Holotype*. One incomplete specimen (MB-A095294) in 96% ethanol, Richards Bay harbour, KwaZulu-Natal, South Africa, 28°47′09.6″S 32°04′55.8″E, coll. J. Kara and R. Kara, 1 September 2019. *Paratypes 1–3*. Three incomplete specimens (MB-A095293, MB-A095510, MB-A095292) in 96% ethanol, Richards Bay harbour, KwaZulu-Natal, South Africa, 28°47′09.6″S 32°04′55.8″E, coll. J. Kara and R. Kara, 1 September 2019. *Non-type material*. Three incomplete specimens (MB-A095295–MB-

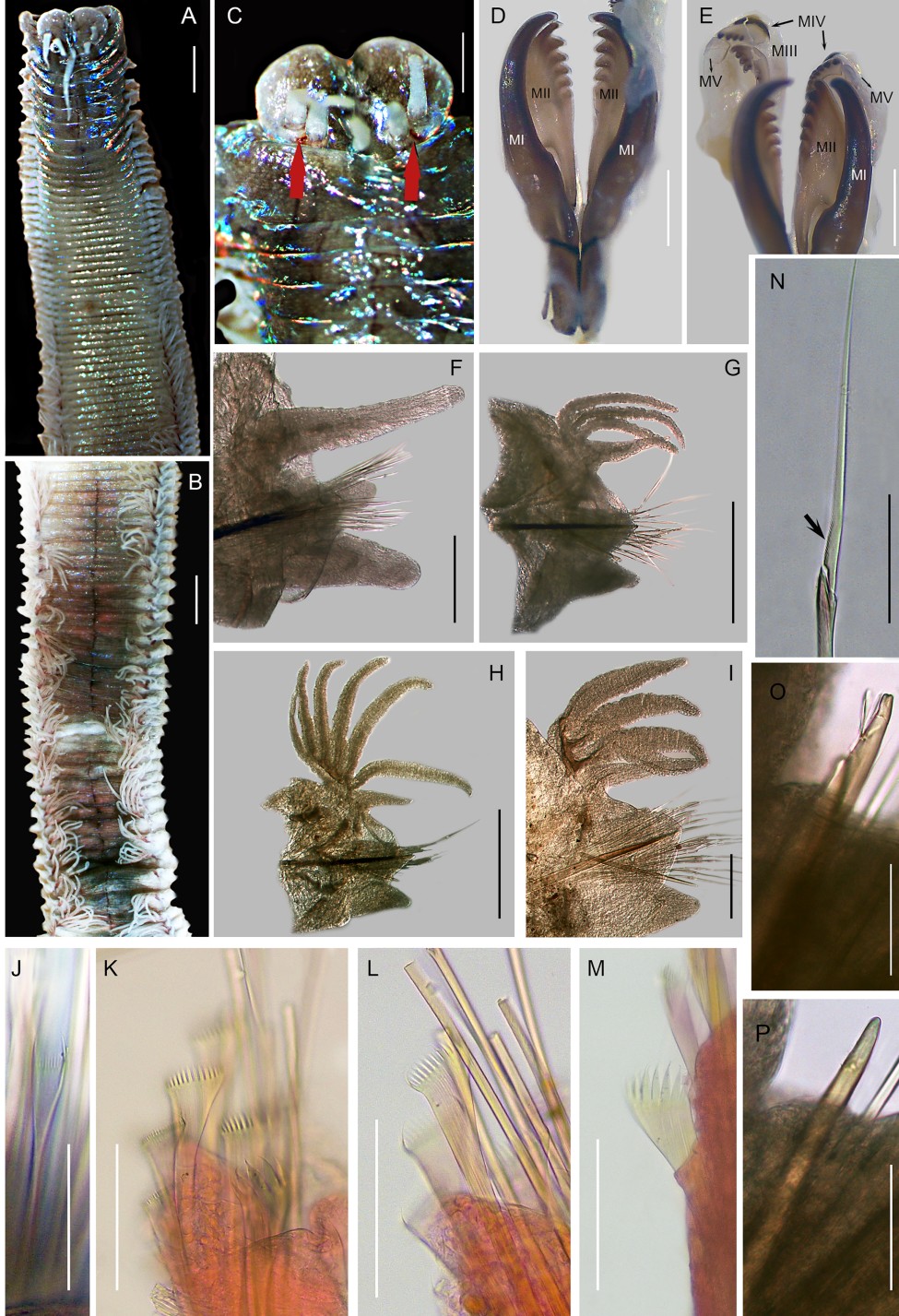

**Figure 4** *Marphysa mzingazia* **sp. nov., MB-A095294.** (A) Anterior region (dorsal), (B) middle region (dorsal), (C) prostomium (dorsal, black arrows = red eyes), (D) maxillary apparatus MI-MII, (E) maxillary apparatus MIII-MV, (F) parapodia from 2ⁿᵈ chaetiger, (G) parapodia from 41ˢᵗ chaetiger, (H) parapodia from 98ᵗʰ chaetiger (dorsal), (I) parapodia from 141ˢᵗ chaetiger (dorsal), (J) thin isodont pectinate chaetae with narrow blade, 13ᵗʰ chaetiger (holotype), (K) thick Isodont pectinate chaetae with wide blade, posterior chaetiger, (L) thick isodont pectinate chaetae with wide blade, posterior chaetiger, (M) thick anodont pectinate chaetae with wide blade, posterior chaetiger, (N) compound spinigers chaeta, 141ˢᵗ chaetiger, (O) subacicular hook, weakly bidentate, (P) subacicular hook, unidentate. Scale bars: A, B = 0.2 mm, C, D = 0.5 mm, E–G = 0.02 mm.

A095297) in 96% ethanol, Richards Bay harbour, KwaZulu-Natal, South Africa, 28°47′09.6″S 32°04′55.8″E, coll. J. Kara and R. Kara, 1 September 2019.

*Description.* Holotype incomplete (posterior end used for molecular analysis), measurements before molecular analysis: 205 chaetigers, L10 = 4 mm, W10 = 3 mm, TL = 55 mm. Body rounded anteriorly, becoming dorsoventrally flattened from chaetiger 7–8 (Figs. 4A–4C) to posterior end, widest at chaetiger 25 (2.49 mm), blood red with white antennae when alive, iridescent throughout, chaetigers 1–8 purple/brown, from segment 9 till posterior parapodia, cream (Figs. 4A and 4B), middle to posterior segments, red midline on segments and pinkish-peach pigment on dorsum (Fig. 4B) when preserved.

Prostomium bilobed (0.71 mm long, 1.48 mm wide), lobes frontally rounded, ventral sulcus deep (Figs. 4A and 4C). Prostomial appendages in semicircle, median antenna slightly isolated by a gap (Figs. 4A and 4C). Palps reaching 2nd peristomial ring, lateral antennae reaching 2nd chaetiger, median antenna reaching 3rd chaetiger. Palpophores and ceratophores thick, short and ring-shaped; palpostyles and ceratostyles thick, rounded with conical ends. Pair of red eyes as subdermal patches between palps and lateral antennae (Fig. 4C, red arrows). Peristomium (1.8 mm long, 5.2 mm wide), first ring two times longer than second ring (Fig. 4A); separation between the two peristomial rings distinct on all sides (Figs. 4A and 4C).

Maxillary apparatus sclerotized, with MF = 1+1, 6+ 6, 4+0, 4+8, 1+1 (Fig. 4D). MI three times longer than maxillary carriers, 3.5 times longer than closing system, forceps-like, falcal arch slightly extended with inner base slightly rounded. MII wide, with triangular teeth directed laterally (Figs. 4D and 4E); three times longer than cavity opening. MIII short, slightly curved, blunt triangular teeth, attachment lamella irregular, situated only in center of anterior edge of maxilla, slightly sclerotized (Fig. 4E). Left MIV with distal tooth longest, attachment lamella semicircle, wide, better developed in central portion, situated 2/3 along anterior edge of maxilla, strongly sclerotized. Right MIV with distal teeth larger; attachment lamella semicircle, wide, better developed in central portion, situated 2/3 along anterior edge of maxilla, slightly sclerotized. MV rectangular, longer than wide, with rounded tooth (Fig. 4E). Mandibles brown with transparent cutting plates.

Palmate branchiae with six short filaments, from chaetigers 18 to posterior end (Figs. 4G–4I). First seven chaetigers with one branchial filament, two in chaetigers 26–28, four in chaetiger 29–59, five in chaetigers 60–90 and a maximum of six filaments from chaetiger 91 to posterior end. Branchial filaments slightly longer than dorsal cirri in anterior chaetigers and double in length in posterior chaetigers.

First three parapodia smallest, best developed in chaetigers 7–31, gradually becoming smaller (Fig. 4A). Dorsal cirri conical in most chaetigers, 1.6 times longer than ventral cirri in anterior, best developed in chaetigers 1–25 (Fig. 4F). Ventral cirri digitiform in first two parapodia (Fig. 4F); with short swollen base and rounded tip from chaetiger 3 to midbody (Fig. 4B); becoming triangular with pointed tip in following chaetigers (Figs. 4G–4I). Prechaetal lobe as a transverse fold in all chaetigers (Figs. 4G–4I). Chaetal lobe conical in anterior chaetigers (Fig. 4F); triangular in middle to posterior chaetigers, with aciculae barely emerging from midline (Figs. 4G–4I). Postchaetal lobe well developed and longer

than chaetal lobe in first 42 chaetigers; digitiform in chaetigers 4–32; ear-shaped in chaetiger 33 to 42, progressively smaller in chaetiger 43 onwards (Figs. 4G–4I).

Aciculae with black shafts and amber blunt tip from anterior to middle chaetigers (Figs. 4F–4H), becoming golden in posterior (Fig. 4I), up to two aciculae in anterior to middle chaetigers, reducing to one in posterior chaetigers (Figs. 4F–4I). Limbate chaetae only supracicular. Four types of pectinate chaetae: (1) 5–6 isodont symmetrical in anterior parapodia, with narrow blades, thin shaft and 11–13 short, slender teeth (Fig. 4J), in anterior region; (2) 5–6 isodont asymmetrical with wide blades, thick shaft, 20–21 short and slender teeth (Fig. 4K), from anterior to middle region; (3) 4–5 anodont, asymmetrical with wide blades, thick shaft 13–14 medium-coarse short teeth (Fig. 4L) in middle to posterior chaetigers; (4) 1–2 anodont with asymmetrical, wide blades, thick shafts, 5–7 long and thick teeth and wide blades (Fig. 4M), in posterior parapodia. Compound spinigers with blades of two lengths in all chaetigers, longer blades more abundant (Fig. 4N). Compound falcigers absent. Subacicular hooks blunt or weakly bidentate; yellow or brown/golden (Figs. 4O and 4P), starting from chaetiger 25, two per parapodium in anterior chaetigers, one in posterior region, present in all parapodia.

**Variation.** Complete specimens with 201 chaetigers, L10 = 3–5 mm, W10 = 3–4 mm, TL = 23–65 mm. Palps reaching from $2^{nd}$ peristomial ring to $3^{rd}$ chaetiger, lateral antenna reaching from 2–4 chaetiger, median antenna up to $3^{rd}$–$4^{th}$ chaetiger. MF varies: MII: 4–5 + 5–6, MIII: 6–7, MIV: 3–4 + 6–8. Start of branchiae in chaetigers 18 to 26. Maximum number of branchial filaments in chaetigers 73–97. Subacicular hooks first occur in chaetigers 23–36. Postchaetal lobe well developed in chaetigers 1–43.

**Habitat.** Found in crevices of muddy rocks (porous rocks made up of fine-grained particles) on the banks of the Mzingazi canal.

**Distribution.** Mzingazi Canal, leading to Richards Bay harbour, KwaZulu-Natal, South Africa.

**DNA barcode.** Richards Bay, KwaZulu-Natal, South Africa. Holotype: MB-A095294, GenBank accession number OQ836470. A total of 575 bp fragment isolated with the universal mitochondrial cytochrome oxidase subunit 1 gene, primer pair LCO1490, HCO2198 (*Folmer et al., 1994*).

**Etymology.** The specific epithet *mzingazia* refers to the type locality *i.e.*, the Mzingazi canal.

**Remarks.** *Marphysa capensis* (Schmarda, 1861), *M. durbanensis* Day, 1934, *M. haemasona* Quatrefages, 1866, *Marphysa sherlockae* Kara, Molina-Acevedo, Zanol, Simon & Idris, 2020, and *M. mzingazia* sp. nov. occur in South Africa and have branchiae throughout the body. *Marphysa capensis* only has compound falcigers in all parapodia, while all others have compound spinigers in addition to the falcigers. *Marphysa durbanensis* and *M. haemasona* have long branchial stems, while these are short in *M. mzingazia* sp. nov. Moreover, *M. haemasona* has colourless eyes (red in *M. mzingazia* sp. nov.) and ovoid

postchaetal lobes in the first four chaetigers (digitiform in *M. mzingazia* sp. nov.). *Marphysa durbanensis* and *M. sherlockae* have reddish subacicular hooks, while these are amber in *M. mzingazia* sp. nov. (*Day, 1967*; *Kara et al., 2020b*).

*Marphysa mzingazia* sp. nov. resembles *M. aransensis* Treadwell, 1939 (Texas), *M. brevibranchiata Treadwell, 1921* (Bahamas), *M. fauchaldi Glasby & Hutchings, 2010* (Australia), *M. gravelyi* Southern, 1921 (India), *M. hongkongensa Wang, Zhang & Qiu, 2018* (China), *M. gaditana* Martin, Gil and Zanol in *Martin et al. (2020)* (Spain), and *M. kristiani Zanol, da Silva & Hutchings, 2016* (Australia) in having only compound spinigers in all chaetigers, and amber subacicular hooks. However, *M. mzingazia* sp. nov. has four types of pectinate chaetae: isodont narrow pectinate with long and slender teeth, isodont wide pectinate with short and slender teeth, anodont wide pectinate with short and slender teeth, anodont wide pectinate with long and thick teeth while *M. aransensis* (isodont narrow pectinate with long and slender teeth, isodont wide pectinate with short and thick teeth, anodont wide pectinate with long and thick teeth), *M. fauchaldi* (isodont narrow pectinate with long and slender teeth, isodont wide pectinate with short and slender teeth, anodont wide pectinate with long and slender teeth), *M. gravelyi* (isodont narrow pectinate with short and slender teeth, isodont wide with long and slender teeth, anodont wide pectinate with long and slender teeth), *M. kristiani* (isodont narrow pectinate with long and slender teeth, isodont wide pectinate with long and slender teeth, anodont wide pectinate with long and thick teeth), and *M. gaditana* (isodont narrow pectinate with long and slender teeth, isodont wide pectinate with long and thick teeth, anodont wide pectinate with long and thick teeth) have only three types of pectinate chaetae, and *M. hongkongensa* has five types of pectinate chaetae (isodont narrow pectinate with long and slender teeth, isodont wide pectinate with short and slender teeth, anodont wide pectinate with long and thick teeth, anodont wide pectinate with long and slender teeth). Furthermore, *M. mzingazia* sp. nov. has an auricular postchaetal lobe from chaetiger 4, whilst in *M. brevibranchiata* the postchaetal lobe is rounded in the same chaetiger. Finally, *M. mzingazia* sp. nov. has conical chaetal lobes in first chaetigers, while in *M. brevibranchiata* the chaetal lobe is rounded in first chaetigers.

### *Treadwellphysa Molina-Acevedo & Carrera-Parra, 2017*

**Diagnosis.** According to *Molina-Acevedo (2019)*: Prostomium bilobed; eyes present; with five prostomial appendages arranged in a semicircle; peristomial cirri absent; branchiae present along the body, long and digitiform branchial filaments. Maxillary apparatus with four pairs (I, II, IV, V) of maxillae and an unpaired (III). Maxillae I forceps-like, attachment lamellae absent; angular shaped falcal arch; base of maxilla with straight outer edge and basal inner with a curvature where the base of MII is supported. Maxillae II, inner base with a small, rounded projection that fits in the curvature of the basal inner edge of maxillae I; similar size in left and right of cavity opening, upper end reaching same height as basal tooth, without attachment lamella. Maxilla III curved; attachment lamella at the centre of posterior edge of maxilla. Maxillae IV; wide attachment lamellae, situated on posterior edge of maxillae. Maxillae V unidentate, attachment lamellae absent. Dorsal cirri

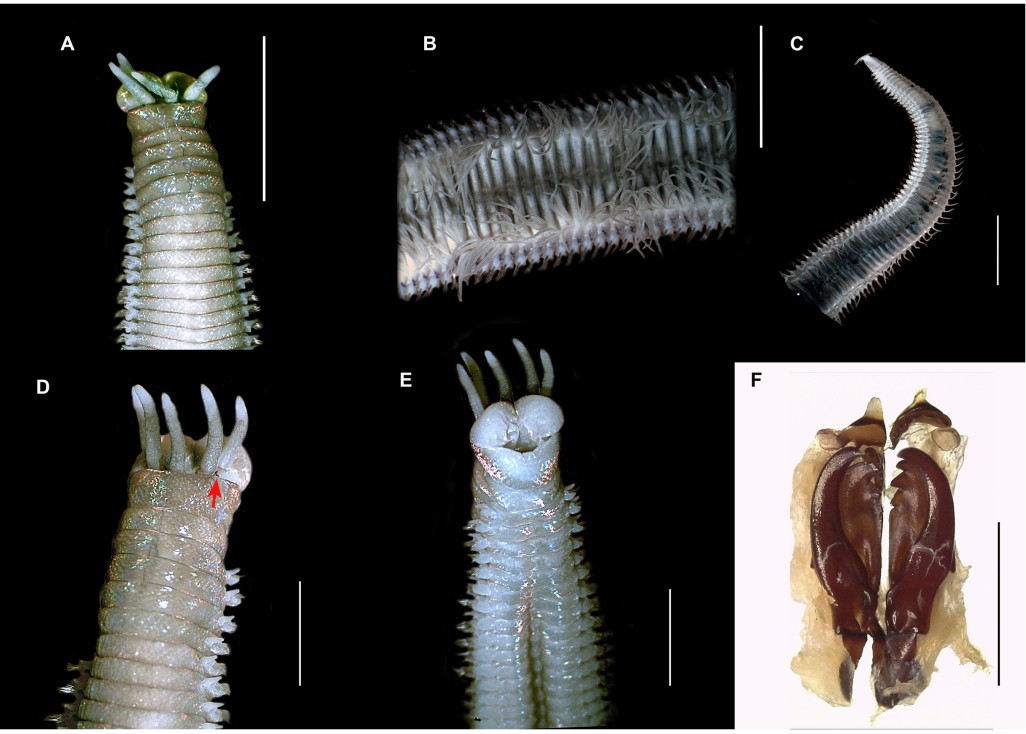

**Figure 5** *Treadwellphysa izinqa* **sp. nov.** (A) Anterior region (dorsal), (B) middle region (dorsal), (C) posterior region (dorsal) (MB-A095291), (D) anterior region (dorsal, red arrow: reniform eye), (E) ventral view (MB-A095512), (F) maxillary apparatus (MB-A095511). Scale bars: A–C = 5 mm, D, E = 2 mm, F = 1 mm.

on parapodia without articulation; gradually decreasing in size in middle toward posterior parapodia; ventral cirri with swollen base as a transverse welt with short digitiform tip, in more than half of parapodia of the body; poorly developed postchaetal lobe. Supracicular chaetae with limbate chaetae; and three shapes of pectinate chaetae, isodont narrow in anterior region, isodont wide in median-posterior region; and anodont wide in posterior region. Subacicular chaetae with compound falcigers always present, bidentate, or tridentate; spinigers, and spinifalcigers (long blade similar to that of a spiniger with bidentate teeth at the proximal end similar to a falciger, but lacks the general "hood" characteristic of falcigers, see *Molina-Acevedo & Carrera-Parra (2017)*) present occasionally; subacicular hooks bidentate present in most parapodia.

***Treadwellphysa izinqa*** **sp. nov.**

lsid:zoobank.org:act:256FCA20-DEEA-463F-BABB-BD7A23D33695

(**Figs. 5–6**)

*Marphysa corallina*—Day 1954:19, *Day, 1967*: 400, fig. 17.7f –j (*Non* Kinberg, 1865).
*Marphysa* cf. *corallina*—Simon, Kara, du Toit, van Rensburg, Naidoo, Matthee 2021:30–31, fig. 15.

***Material examined****. Holotype.* One incomplete specimen (MB-A095291) in 96% ethanol, Umhlanga Rocks, KwaZulu-Natal, South Africa, 29°43′39.2′′S 31°05′20.0′′E, coll. J. Kara,

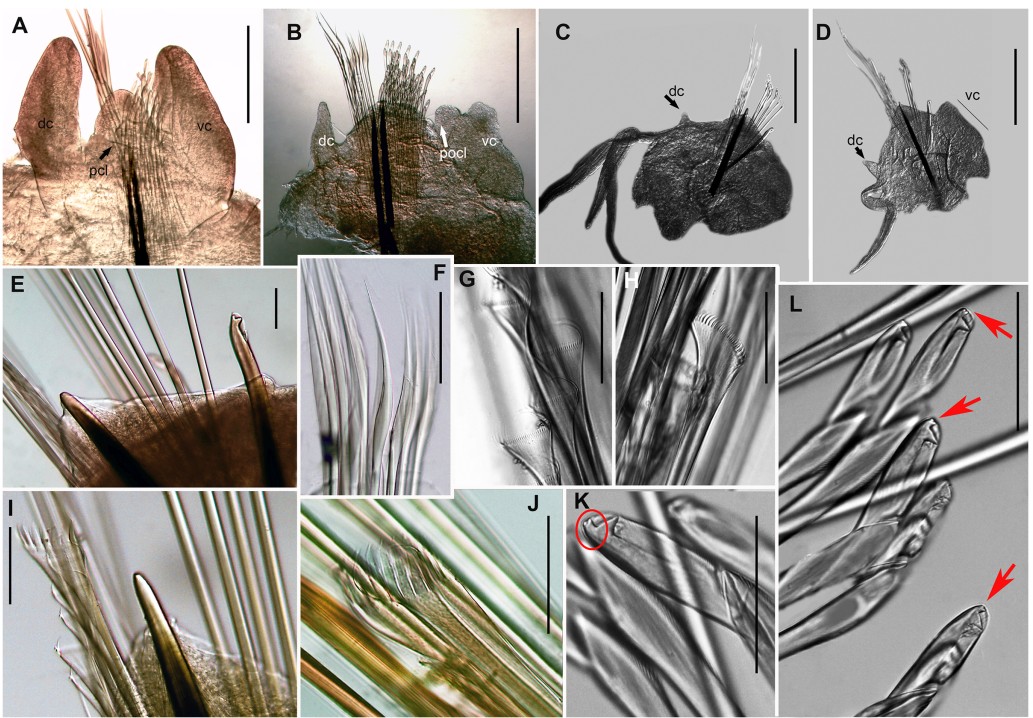

**Figure 6** *Treadwellphysa izinqa* sp. nov., MB-A095290. **Parapodial and chaetal morphology.** (A) Parapodia 2nd chaetiger, (B) parapodia 15th chaetiger, (C) parapodia 56th chaetiger, (D) parapodia posterior chaetiger, (E) aciculum and bidentate subacicular hook. (F) Limbate capillaries. (G) Thin isodont pectinate chaetae, wide with fine short teeth, (H) thick anodont pectinate chaetae, wide with short fine teeth, posterior chaetiger, (I) thick anodont pectinate chaetae, wide with thick short teeth, posterior chaetiger, (J) thick anodont pectinate chaetae, wide with short thick teeth, posterior chaetiger, (K and L) compound tridentate falcigers chaetae, posterior chaetiger. Parapodia are in dorsal view. Scale bars: A = 0.2 mm, B–D = 0.5 mm, E–L = 0.05 mm.

19 September 2019. *Paratypes 1–4.* Four incomplete specimens (MB-A095512, MB-A095511, MB-A095290, MB-A095287) in 96% ethanol, Umhlanga Rocks, KwaZulu-Natal, South Africa, 29°43′39.2″S 31°05′20.0″E, coll. J. Kara, 19 September 2019. *Non-type material.* Three specimens, two incomplete, one complete (SAMC-A20577), Umhlali Shore station, KwaZulu-Natal, South Africa, det. J. H. Day. Nine specimens (MB-A095280–MB-A095289) in 96% ethanol, Umhlanga Rocks, KwaZulu-Natal, South Africa, 29°43′39.2″S 31°05′20.0″E, coll. J. Kara, 19 September 2019. Fourteen specimens (MB-A095266–MB-A095279) in 96% ethanol, Port Shepstone, KwaZulu-Natal, South Africa, 30°44′27.4″S 30°27′34.5″E, coll. J. Kara, 30 September 2019. Five specimens (MB-A095537–MB-A095541) in 96% ethanol, Mabibi, KwaZulu-Natal, South Africa, 27°24′58.3″S 32°42′43.8″E, coll. J. Kara, 29 April 2014. Six specimens (MB-A095517–MB-A095522) in 96% ethanol, Adlams, KwaZulu-Natal, South Africa, 27°37′28.3″S 32°39′22.5″E, coll. J. Kara, 30 March 2014. Four specimens (MB-A095513–MB-A095516) in 96% ethanol, Ballito, KwaZulu-Natal, South Africa, 29°32′23.2″S 31°13′25.9″E, coll. J. Kara, 31 January 2014. Six specimens (MB-A095527–MB-A0995532) in 96 ethanol, Reunion Rocks, KwaZulu-Natal, South Africa, 29°59′11.5″S 30°57′51.0″E, coll. J.Kara, 13 June 2013. Four specimens (MB-A095533–MB-A095536) in 96% ethanol, Green Point,

KwaZulu-Natal, South Africa, 30°15′00.6″S 30°46′55.9″E, coll. J. Kara, 23 June 2013. Four specimens (MB-A095523–MB-A095526), in 96% ethanol, Umgazana, KwaZulu-Natal, South Africa, 31°42′19.3″S 29°24′49.2″E, coll. J. Kara, 10 July 2013. Five specimens (MB-A090276–MB-A090280), in 96% ethanol, Witsand, Western Cape, South Africa, 34°23′31.9″S 20°51′50.1″E, coll. A. du Toit, 30 April 2017.

**Description.** Holotype incomplete (posterior end used for molecular analyses) with up to 151 chaetigers, L10 = 9 mm, W10 = 7 mm, TL = 80 mm. Transversal body section rounded anteriorly, dorsoventrally flattened from chaetiger 18 to posterior end (Figs. 5A–5C); widest at chaetiger 35–40 (0.35 mm). Body light brown anteriorly (Figs. 5A and 5D), cream from middle to posterior end (Figs. 5B and 5C), iridescent throughout. Palps, lateral and median antennae olive green/dark brown with white conical tips (Figs. 5A, 5D and 5E).

Prostomium bilobed (0.1 mm long, 0.3 mm wide), lobes frontally rounded, ventral sulcus deep (Figs. 5A, 5D and 5E). Prostomial appendages in semicircle (Fig. 5D), median antenna slightly isolated by a gap. Palps reaching second peristomial ring, lateral and median antennae reaching chaetigers 1 and 2, respectively. Palpophores and ceratophores thick, short, ring shaped; palpostyles and ceratostyles thick, cylindrical with conical ends. Two black ring-shaped eyes at the base of lateral antennae (Fig. 5A, red arrow).

Peristomium (0.1 mm long, 0.25 mm wide) with first ring twice as long as second, clearly separated on all sides (Figs. 5A, 5D and 5E).

Maxillary apparatus sclerotized, MF = 1+1, 3+3, 4+0, 3+5, 1+1 (Fig. 3F). MI 2.4 times longer than maxillary carriers, three times longer than closing system, MI forceps-like, slightly extending but rounded, with curved basal inner edge (Fig. 5F). MII wide, with triangular teeth directed laterally, three times longer than cavity opening, and a small elevation at base, fitting inner edge of maxillae I (Fig. 5F). MIII short, with blunt triangular teeth and irregular attachment lamella at centre of posterior edge in relation to maxilla (Fig. 5F). Right MIV with distal tooth longest, and strongly sclerotized rectangular attachment lamella at 2/3 along posterior edge of maxilla. Right MIV with three teeth, distal tooth longest, and sclerotized circular attachment lamella along posterior edge of maxilla. MV rectangular, square, with single tooth (Fig. 5F). Mandibles brown with transparent cutting plates and six growth rings.

Branchiae pectinate, with up to three long filaments, starting from chaetiger 26 to posterior end (Fig. 6C); with a single filament at chaetigers 26 to 37, three long filaments from chaetigers 38–97, two filaments from chaetigers 98–120, and one filament from chaetigers 121 to posterior end (Figs. 5B, 5C, 6C and 6D). Branchial filaments nine times longer than dorsal cirri in median body region (Figs. 6C and 6D).

First three parapodia smallest (Figs. 6A, 6D and 6E). Dorsal cirri digitiform in anterior chaetigers, two times longer than postchaetal lobes (Fig. 6A); conical in median-posterior chaetigers, 1/2 length of chaetal lobe (Fig. 6C) and 1/3 longer than chaetal lobe in posterior chaetigers (Fig. 6D). Ventral cirri digitiform in first three parapodia, 2.3 times longer than postchaetal lobes (Fig. 6A); with oval swollen base and rounded tip from chaetiger 4–8; with transverse welt with a rounded tip from chaetiger 9–97; reducing to small swollen base with round tip from chaetiger 98 to posterior (Figs. 6B–6D). Prechaetal lobe as a

transverse fold in all chaetigers (Figs. 6A–6D). Chaetal lobe rounded in anterior chaetigers (Figs. 6A and 6B); triangular in median to posterior chaetigers, with aciculae emerging from midline (Figs. 6C–6E). Postchaetal lobe developed in first 38 chaetigers and longer than chaetal lobe; digitiform in first two chaetigers; ear-shaped from chaetigers 30–37 progressively smaller and inconspicuous from chaetiger 38 to posterior end (Figs. 6B–6E).

Aciculae with black shafts and amber blunt tip (Figs. 6E and 6I), two aciculae from anterior to median chaetigers, one in posterior chaetigers (Figs. 6A–6E and 6I). Limbate capillaries of two lengths in supracicular position in all chaetigers (Fig. 6F). Four types of pectinate chaetae; in anterior chaetigers, 4–6 isodont symmetrical pectinate chaetae, with wide blade, thin shaft, up to 26–28 short and slender teeth (Fig. 6G); in median to posterior chaetigers, 1–2 isodont asymmetrical pectinate chaetae, with wide blade, thick shaft and up to 21 long and slender teeth (Fig. 6H), 1–4 anodont pectinate chaetae, with wide blade, thick shaft and up to 8–10 short and thick teeth (Fig. 6I), 1–2 anodont pectinate chaetae, with wide blade, thick shaft and up to 8–10 long and thick teeth (Fig. 6J). Compound spinigers absent. Compound falcigers tridentate with a triangular proximal tooth larger than rounded distal teeth in anterior parapodia; in median to posterior chaetigers, with blades of similar length, shorter than anterior falcigers, with rounded distal teeth shorter than triangular proximal tooth (Figs. 6K and 6L). Subacicular hook bidentate with rounded guard, reddish basally and golden at distal end; teeth triangular with proximal tooth larger than distal tooth (Fig. 6E); starting from chaetiger 28–30, usually one per parapodium, present in all chaetigers.

Pygidium with two pairs of pygidial cirri emerging ventrally, dorsal pair as long as last five chaetigers; ventral pair 1/3 as long as dorsal pair (Fig. 5C).

*Variation*. Paratypes incomplete, with up to 145–176 chaetigers, L10 = 7–8 mm, W10 = 5–6 mm and TL = 75–88 mm. Palps reaching between first and second peristomial rings. Lateral and median antenna reaching between chaetiger 2 and 3. Non-type material: L10 = 0.6–1.5 mm, W10 = 0.2–0.55 mm; palps reaching between first peristomial ring and chaetiger 2, lateral antenna reaching between the first peristomial ring and chaetiger 3, median antenna reaching between first peristomial ring and chaetiger 5. Branchiae starting from chaetiger 22–42, maximum number of branchial filaments from chaetiger 49–55. Subacicular hooks starting from the chaetiger 25 to 30. Postchaetal lobe well-developed from chaetiger 1–38. Maxilla formula: MII 2+3, MIV 3+5.

*Habitat*. Found in mucous-sand burrows in sediment under coralline algal beds, worm rock (dense aggregation of calcium carbonate-sand tubes created by sabellarid and/or serpulid worms, forming a rock-like structure), and in crevices on fringing intertidal rocky shores.

*Distribution*. Northern KwaZulu-Natal (Mabibi) to Eastern Cape (Umgazana), Witsand in the Western Cape (*Simon et al., 2021b*).

*DNA barcode*. Umhlanga Rocks, KwaZulu-Nata, South Africa. Holotype MB-A095291, GenBank accession number OQ836467. A total of 575 bp fragment isolated with the

universal mitochondrial cytochrome oxidase subunit 1 gene, primer pair LCO1490, HCO2198 (*Folmer et al., 1994*).

***Etymology.*** The isiZulu (the native language of the KwaZulu-Natal people) word "izinqa" translates to buttocks and refers to the round prostomial lobes that are separated by a deep ventral sulcus, giving a distinctive butt-like appearance that can be seen with the naked eye.

***Common name.*** Brown wonderworm. Live *T. izinqa* sp. nov. can be distinguished from *M. haemasona*, the blood wonderworm commonly used as bait in the Western Cape, in having a solid brown body colour and white tipped antennae and palps (see fig. 15A in *Simon et al., 2021b*), whereas *M. haemasona* has a reddish/deep violet, speckled anterior dorsum, and antennae tips with brown and white bands (*Simon et al., 2022*).

***Remarks.*** *Marphysa* cf. *corallina* from Witsand (*Simon et al., 2021b*) matches the description of *T. izinqa* sp. nov., particularly in having a small elevation at the inner base of maxilla II fitting in the inner curved base of MI but with a few minor differences that are probably site related. Specimens from Witsand are 40 mm longer than the specimens examined here; have two to four more branchial filaments and branchiae that start between 9–13 chaetigers later. However, the low intraspecific distances (<1%) of specimens collected from Witsand and KwaZulu-Natal confirms that they belong to the same species. Moreover, morphology of the maxillary apparatus, also considered a stable character (*Molina-Acevedo & Carrera-Parra, 2017*), conform to that of *T. izinqa* sp. nov. with having the small elevation at the inner base of MII fitting in the inner, curved base of MI (fig. 15F, *Simon et al., 2021b*). Thus, specimens from Witsand are considered here as *Treadwellphysa izinqa* sp. nov.

Treadwellphysa izinqa sp. nov. resembles *T. rizzoae Molina-Acevedo, 2019*, *T. villalobosi Molina-Acevedo, 2019*, *T. languida* (*Treadwell, 1921*), and *T. veracruzensis* (*De León-González & Castañeda, 2006*) in having only compound falcigers throughout the body. However, *T. villalobosi*, *T. languida* and *T. veracruzensis* have translucent subacicular hooks, and compound falcigers with blades of various lengths in anterior region, whereas in *T. izinqa* sp. nov. the subacicular hook is reddish basally and distally translucent, and falcigers are present with consistent length in anterior chaetigers. *Treadwellphysa rizzoae* has falcigers in anterior chaetigers only with similar lengths, and a poorly developed postchaetal lobe which is rounded in most chaetigers when present, while *T. izinqa*. sp. nov. has tridentate falcigers, well-developed postchaetal lobes that are digitiform in first chaetigers and auricular in following ones.

## Identification key for the *Marphysa* species from South Africa

1 With only compound falcigers.......................... *M. capensis* (Schmarda, 1861)
- With compound falcigers and spinigers in anterior-median region. *M. sherlockae* Kara, Molina-Acevedo, Zanol, Simon & Idris, 2020
- With only compound spinigers .................................................. 2

2(1) Palmate branchiae, with short filaments, exceeding the length of the dorsal cirri ............................................................. *M. mzingazia* sp. nov.

- Pectinate branchiae, with long filaments, at least four times longer than dorsal cirri in median region of the body............................................................. 3

3(2) Postchaetal lobe digitiform in first three chaetigers, MII 5–6+6–8, and five types of pectinate chaetae: narrow isodont pectinate with long and slender teeth, wide isodont pectinate with short and slender teeth, wide isodont pectinate with short and thick teeth, wide anodont pectinate with short and slender teeth, and wide anodont with long and thick teeth ............................................... *M. durbanensis* Day, 1934

- Postchaetal lobe tongue-shaped (ovoid) in first three chaetigers, MII 4+4, and four types of pectinate chaetae: narrow isodont pectinate with long and slender teeth, wide isodont pectinate with short and slender teeth, wide anodont pectinate with short and slender teeth, and wide anodont with long and thick teeth............... *M. haemasona* Quatrefages, 1866.

## DISCUSSION

Our study revealed that *M. corallina* was historically misidentified and does not occur in South Africa. Instead, it corresponds to a new endemic species described here as *Treadwellphysa izinqa* sp. nov. and represents the first record of this genus in the country. We also describe a second endemic species, *Marphysa mzingazia* sp. nov. from Richards Bay. As such, only endemic species within this genus occur in South Africa instead of cosmopolitan species as previously reported. This highlights the importance of conducting thorough taxonomic revisions, even in regions with supposedly well-resolved polychaete fauna (*Griffiths et al., 2010*; *Hutchings & Lavesque, 2020*; *Simon et al., 2022*).

The extensive revision of *Marphysa* species has revealed additional key characters for species delimitation, including the maxillary apparatus, variation of chaetae along the body, the shape of parapodial lobes and ventral cirri and the number and type of pectinate chaetae (*Carrera-Parra & Salazar-Vallejo, 1998*; *Zanol, Halanych & Fauchald, 2014*; *Molina-Acevedo & Carrera-Parra, 2017*; *Zanol, da Silva & Hutchings, 2017*). Particularly, the shape of the maxillary apparatus and ventral cirri allowed us to place the South African specimens previously identified as *M. corallina* in the genus *Treadwellphysa*. We also uncovered an additional chaetal type for this genus that was not reported previously *i.e.*, tridentate falcigers.

The shape of the branchiae and postchaetal lobes, the types of pectinate chaetae, and the colour of subacicular hooks were key in identifying the other new species described which clearly belongs to *Marphysa*.

The species-rich subgroup *sanguinea* within *Marphysa* only has compound spinigers, while showing a low morphological complexity and having a low number of distinctive and stable characters which can change with growth (*Molina-Acevedo & Carrera-Parra, 2015*; *Molina-Acevedo & Idris, 2020*). Nonetheless, different combinations of characters produce a unique species-specific morphological pattern which aid in species delimitation.

Molecular data also contribute to unambiguous species delimitation as demonstrated by the two distinct clades reconstructed in our phylogenetic analyses for the two new species described. Similarly, molecular data was useful in identifying the alien species, *Marphysa victori Lavesque et al., 2017* that was mistaken for a new indigenous species in the invaded range, France, helping to stabilise the species name even when large morphological variation was detected within the species (*Lavesque et al., 2020*). On the other hand, two sympatric Iberian species, *M. gaditana* and *M. chirigota* Martin, Gil & Zanol 2020 were initially traded as fish bait under the name "*M. sanguinea*", nonetheless, when revised, morphological and molecular data revealed they were different species (*Martin et al., 2020*).

Our results show that *T. izinqa* sp. nov. occurs widely across South Africa, from the tropical Delagoa inshore ecozone, the subtropical Natal inshore ecozone and the Agulhas inshore ecozone. However, the significantly low intraspecific variation indicates that they are a single, panmictic population. Interestingly, *M. haemasona* was dominant along the cool-temperate west coast (*Simon et al., 2021b*; J Kara, 2023, unpublished data), whereas *T. izinqa* prefers the warm east coast (*Kara, 2015*), suggesting that both species could be limited by temperature. This has been shown for *Platynereis entshonae Kara et al., 2020a* as it was abundant along the cooler west coast whereas *Platynereis* sp. seemed to prefer the warmer southeast coast (*Kara et al., 2020a*). It will be interesting to explore the southeastern coast between Witsand and Umgazana, to be able to untangle the possible presence of *T. izinqa*.

Correctly identifying species exploited as bait is key to understanding their patterns of usage, to develop appropriate management practices and to prevent over-exploitation (*Simon et al., 2021b*). The common name wonderworm, for instance, is widely used for different species of Eunicidae, notably *Marphysa*, *Eunice* and *Lysidice*. Thus, qualifying names were recommended to distinguish between species (*Simon et al., 2021b*, *2022*). Accordingly, as *T. zinqa* sp. nov. has a characteristically brown body and white-tipped antenna we herein propose the common name 'brown wonderworm'. This will allow management authorities to distinguish it from *M. haemasona*, known as 'blood wonderworm' (*Simon et al., 2022*), which has a reddish, deep purple colour with white flecks all over its body and white and brown tipped antennae, when inspecting bait harvested by fishermen. This will facilitate a better understanding of the species that are commonly used throughout its distribution range, understand how harvesting pressures impact the populations along the coast and thus improve management practices (*Simon et al., 2021b*).

In summary, our morphological and molecular data contribute to (1) confirming the previously underestimated endemic diversity of *Marphysa* (*Kara et al., 2020b*) and supporting the finding by *Simon et al. (2022)*, that more than 500 species are unresolved cosmopolitan taxa hiding undescribed local species; (2) understanding the diversity, distribution and biogeography of the endemic *Marphysa*, a genus that was ranked the 7$^{th}$ most important genus requiring revision among the species included in *Day (1967)* monograph (*Simon et al., 2022*), (3) providing fundamental knowledge on generic character stability within Eunicidae, highlighting that morphology of the maxillary

apparatus, chaetae along the body, and parapodial features all have proven reliable at teasing apart species (*Molina-Acevedo & Idris, 2021*; *Capa & Hutchings, 2021*; *Zanol et al., 2021*).

## ACKNOWLEDGEMENTS

We sincerely thank Albe Bosman at the Iziko South African Museum, Cape Town and Lena Gustavsson at the Swedish Museum of Natural History for providing material for examination. We acknowledge the Marine Evolution Laboratory at the University of KwaZulu-Natal for legwork, field and laboratory support (and awesomeness). To Dr. Luis F. Carrera-Parra (El Colegio de la Frontera Sur, Departamento de Sistemática y Ecología Acuática. Estructura y Función del Bentos, Chetumal, Quintana Roo, México) and Dr Sergio I Salazar-Vallejo (El Colegio de la Frontera Sur, Departamento de Sistemática y Ecología Acuática. Estructura y Función del Bentos, Chetumal, Quintana Roo, México) for their advice and conversations on morphology of the Eunicidae family. To Biol Humberto Bahena-Basave (El Colegio de la Frontera Sur, Chetumal, Quintana Roo, México) for advice on digital photography and editing. We also thank editors and reviewers whose comments made a significant contribution to improving this final version.

### Funding

Jyothi Kara was supported by a Postdoctoral Fellowship from the National Research Foundation (Grant Number: 116654) and Isabel C. Molina-Acevedo was supported by a scholarship from CONACyT (514117/298079). The Iziko South African Museums funded the publication fees for this article. The funders had no role in study design, data collection and analysis, decision to publish, or preparation of the manuscript.

### Grant Disclosures

The following grant information was disclosed by the authors:
Postdoctoral Fellowship from the National Research Foundation: 116654.
CONACyT: 514117/298079.
The Iziko South African Museums funded the publication fees for this article.

### Competing Interests

The authors declare that they have no competing interests.

### Author Contributions

- Jyothi Kara conceived and designed the experiments, performed the experiments, analyzed the data, prepared figures and/or tables, authored or reviewed drafts of the article, and approved the final draft.
- Isabel C. Molina-Acevedo conceived and designed the experiments, performed the experiments, analyzed the data, prepared figures and/or tables, authored or reviewed drafts of the article, and approved the final draft.

- Angus Macdonald analyzed the data, authored or reviewed drafts of the article, contributed costs for field work and sequencing, and approved the final draft.
- Joana Zanol analyzed the data, authored or reviewed drafts of the article, and approved the final draft.
- Carol Simon analyzed the data, authored or reviewed drafts of the article, contributed costs for field work and sequencing, and approved the final draft.

### Field Study Permissions

The following information was supplied relating to field study approvals (*i.e.*, approving body and any reference numbers):

The collection of specimens was approved by the Department of Forestry, Fisheries and the Environment in South Africa under permit numbers RES2013/13, RES2014/06 issued to Angus Macdonald and RES2019/49 issued to Carol Simon.

### DNA Deposition

The following information was supplied regarding the deposition of DNA sequences:

The newly generated sequences in this study are available at Genbank: OQ836443–OQ836473.

The specimens used in this study were deposited at the Iziko South African Museums: MB-A095266–MB-A095297.

### Data Availability

The sequences and spreadsheet with museum and Genbank accession numbers are available in the Supplemental Files.

### New Species Registration

The following information was supplied regarding the registration of a newly described species:

Publication LSID: urn:lsid:zoobank.org:pub:091D2E19-B708-4FAA-9E10-4BC0EA8E28AA

Treadwellphysa izinqa LSID: urn:lsid:zoobank.org:act:256FCA20-DEEA-463F-BABB-BD7A23D33695.

Marphysa mzingazia LSID: urn:lsid:zoobank.org:act:6A50F2CF-2DE3-42AC-8B4D-FECC6C04D4BF.

### Supplemental Information

Supplemental information for this article can be found online at http://dx.doi.org/10.7717/peerj.16665#supplemental-information.

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
