# Peer review of "A closer look at the taxonomic and genetic diversity of endemic South African Marphysa Quatrefages, 1865"

_PeerJ, doi:10.7717/peerj.16665_

## Round 0.1 · original submission · Major Revisions

The manuscript interestingly described the last remaining unresolved cosmopolitan species of the genus Marphysa in South Africa. I agree with the points/concerns made by the reviewers and would like to draw the author's attention to address the suggestions raised in the two annotated pdf uploaded by both reviewers. I look forward to receiving the revised manuscript soon.

Reviewer 1 ·

Basic reporting

Article is well generally well written have made some suggestions throughout

Experimental design

not that relevant to this study

Validity of the findings

2 new species are well described

Additional comments

some carefully editing re punctuation is required have marked on the manuscript

Annotated reviews are not available for download in order to protect the identity of reviewers who chose to remain anonymous.

·

Basic reporting

The manuscript requires revision in nearly all concepts targeted in this section, except for being self-contained and include relevant results (i.e., english style, literature references, figures and tables require revision). However, the list of suggestions and comments is too long to be here detailed. All them can be found in the attached revised manuscript.

Experimental design

Do not apply.

Validity of the findings

The results of this manuscript contain very interesting information on the species of Marphysa and related genus in southern Africa, which include the description of two new species from the region, leading to having a fauna with indigenous species instead of worldwide cosmopolitan taxa, as previously stated. The findings are well supportted both in terms of morphological and molecular data, and I have no concerns on the validity of the newly described species. So, I strongly recommend to publish this manuscript, but not before without writing an accuratly revised new version.

Additional comments

An additional concern, besides all comments and sugestions included in the attached manuscript, is that there are many concepts repeated in slightly different forms in defferent sections of the manuscript. I strongly recommend the authors to critically review my sugestions/comments, but also to read carefully the new version they will produce trying to avoid these repeated parts of the text. I also recommend them to check for consistency all along the manuscript, and to check if the use of abbreviations is or not necessary. In case of being considered as necessary, these have to be explained in the text, preferably in the methods section. I also strongly recomment them to try to improve the quality of their photos and, particularly in the case of the pectinate chaetae, if they do not have better photos, to provide drawings of all types for all species they are describing, which will be very useful for future readers.

---

## Round 0.2 · Major Revisions

I hope the authors will consider revising the manuscript and providing a point-to-point rebuttal when resubmitting.

**Language Note:** The review process has identified that the English language must be improved. PeerJ can provide language editing services - please contact us at copyediting@peerj.com for pricing (be sure to provide your manuscript number and title). Alternatively, you should make your own arrangements to improve the language quality and provide details in your response letter. – PeerJ Staff

Reviewer 1 ·

Basic reporting

The Introduction and Discussion still need a lot of work- as long convoluted sentances - have highlighted where major rewrites are required
and references are not in a consistent format

some of the authors are fluent in English they need to carefully check the final manuscript

Experimental design

no comment

Validity of the findings

they just to better present their conclusions- and restrict themselves to their findings - and the Intro needs a major reorganisatiom have no problems with their descriptions of species -

Additional comments

please see my comments on the attached manuscript

major rewrite of Intro and Discussion needed and the format of the refs not consistent

Annotated reviews are not available for download in order to protect the identity of reviewers who chose to remain anonymous.

---

## Round 0.3 · Minor Revisions

Dear authors, I welcome you to re-submit your manuscript after addressing some minor comments.

Reviewer 1 ·

Basic reporting

no comment

Experimental design

No comment

Validity of the findings

no comment

Additional comments

some minor comments - typos and formatting for refs needs checking
see attached file

Annotated reviews are not available for download in order to protect the identity of reviewers who chose to remain anonymous.

---

## Round 0.4 · accepted · Accept

The paper is ready for publication.